# Preparation of Antimicrobial Fibres from the EVOH/EPC Blend Containing Silver Nanoparticles

**DOI:** 10.3390/polym12081827

**Published:** 2020-08-14

**Authors:** Dorota Biniaś, Włodzimierz Biniaś, Alicja Machnicka, Monika Hanus

**Affiliations:** 1Institute of Textile Engineering and Polymer Materials, University of Bielsko-Biala, Willowa 2, 43-309 Bielsko-Biala, Poland; wbinias@ath.bielsko.pl; 2Institute of Environmental Protection and Engineering, University of Bielsko-Biala, Willowa 2, 43-309 Bielsko-Biala, Poland; amachnicka@ath.bielsko.pl (A.M.); monikahanus235@gmail.com (M.H.)

**Keywords:** EVOH, EPC, microporus fibres, silver nanoparticles, antibacterial activity

## Abstract

The article presents a new fabrication method for bioactive fibres with a microporous structure of ethylene–vinyl alcohol copolymers (EVOH)/ethylene−propylene copolymer (EPC) blends. The experimental work carried out resulted in obtaining EVOH/EPC polymer blends fibres with the addition of glycerol and sodium stearate. Different concentrations of glycerol (38%, 32%) and sodium stearate (2%, 8%) were used to prepare the fibres. The purpose of using different concentrations of stearate and glycerol was to evaluate the effect of additives on the structure and properties of the fibres. A significant influence of the additives used on the morphological structure of the fibres was found. The resulting fibres were modified with an AgNO_3_ solution and reduced to silver nanoparticles (AgNPs), to give the fibres bioactive properties. The fibres obtained with the addition of 8% stearate have a more developed surface, which may influence the amount of adsorbed silver particles inside the fibre. However, the durability of depositing silver particles after multiple washes has not been tested. Three types of microorganisms were selected to assess the microbiological activity of the obtained fibres, i.e., Gram-positive *Staphylococcus aureus* and Gram-negative *Pseudomonas aeruginosa* and *Escherichia coli*. The fibres have antibacterial activity against gram positive and negative bacteria. The largest inhibition zones were obtained for gram-positive bacteria *Staphylococcus aureus*, ranging from 3 to 10 mm depending on the concentration of AgNPs. The morphology of the blends fibres was characterized by scanning electron microscopy (SEM) and optical microscopy (OM). The occurrence of elemental silver was analysed by energy dispersive spectroscopy (EDS) analysis. The changes of the polymer structure chemistry are studied by Fourier transform infrared spectroscopy (FTIR).

## 1. Introduction

Ethylene–vinyl alcohol copolymers (EVOH) are constructed by random sequencing of hydrophobic ethylene and hydrophilic vinyl alcohol monomeric units. The EVOH properties change substantially with the change in content of individual mer units. EVOH belongs to hygroscopic and water absorbing polymers. This effect has been studied [1,2,3]. It has good thermal stability in processing and high chemical resistance. EVOH is used for barrier materials with low gas permeability, and is mainly applied in the automotive industry and food packing. This material is used in a wide variety of applications [4,5,6,7,8,9]. An ethylene vinyl alcohol copolymer (EVOH), with high mechanical strength and good biocompatibility, was used as the polymeric materials. It has high mechanical strength in the form of foils and fibres [10,11,12,13]. Moreover, it is characterized by good biocompatibility. This has been studied [14,15,16,17,18].

The thermoplastic elastomer (TPE), based on propylene-dominant ethylene−propylene copolymer (EPC), was studied. The ethylene-propylene copolymers (EPC) commercial grade Vistamaxx™ 2120 is distinguished by good elasticity properties and can be processed along with many polymers, including polyolefins, improving their flexibility. This has been studied [19,20,21].

Silver nanoparticles (AgNPs) are some of the most commercialized nanomaterials, extensively used as fungicides or bactericides in diverse applications, such as personal care products, food, clothing, building materials and medical equipment. This has been studied [22,23,24]. There are various synthetic methods used for the preparation of AgNPs. This result was later contradicted by Kumar et al. [25] AgNPs is a well-known nanomaterial with broad-spectrum antibacterial effects on viruses, fungi, Gram-negative and Gram-positive bacteria [26,27,28,29]. Thus, intensive research was focused on the preparation of AgNPs, using safe natural polymers and other polymers, textile and food packing [30,31,32,33,34,35].

Silver ions are characterized by a high reactivity to bacteria. In numerous studies, the affinity of Ag^+^ ions to phosphate, carboxyl and amino functional groups was found. Silver ions get through the bacterial shield and bind to the phospholipid layer of the membrane, which leads to its disruption and destruction of bacteria as a result of improper transport of ions and metabolites. In addition, bacterial deactivation also takes place by catalytic oxidation. The surface of silver ions favours the absorption of oxygen, and this causes the oxidation of –S–H thiol groups of bacteria, which destroys the so-called bacterial respiratory chain [36].

Among the non-degradable polymers, ethylene vinyl alcohol (EVOH) is the most largely used to host AgNPs for food packaging. In the work of Carbone M et al. [37], silver ions were incorporated into an EVOH polymer matrix and the antibacterial efficacy was tested in contact with different kinds of food.

## 2. Materials and Experimental Methods

### 2.1. Materials

The ethylene–vinyl alcohol (EVOH), commercial grade F101B, with an ethylene content of 32%, was obtained from Kuraray (Japan). The ethylene-propylene copolymers (EPC) commercial grade Vistamaxx™2120 based olefinic elastomer is produced using Exxon Mobil Chemical’s Exxpol™ technology (Spring, TX, USA).

Glycerol (C_3_H_8_O_3_), sodium stearate (C_18_H_35_NaO_2_) and silver nitrate (AgNO_3_) were supplied by Avantor Performance Materials Poland S.A., and used directly without further purification.

Some of the properties of EVOH and EPC are listed in Table 1.

The chemical structure of components is shown in Figure 1.

### 2.2. Preparation of Fibre Blends

Before melt blending, EVOH and EPC were dried at 60 °C for 4 h. All of the components, i.e., EVOH, EPC, glycerol and sodium stearate, were mixed in appropriate proportions. EPC and EVOH granules were mixed with sodium stearate hot-dissolved in glycerol.

The fibres were formed on a Zamak Mercator (Poland) twin screw extruder. The mixture was placed in a hopper of the extruder. The fibres were formed using a single screw extruder at 160 °C. A single-hole spinneret with a diameter of 0.25 mm was used. The fibre stretch ratio was 200%, and the rate of fibre collection was 500 m/min. Table 2 shows the composition of the different blends.

The glycerol used is an important component of the alloy because it dissolves sodium stearate. It gives the fibres a porous character because the sodium stearate washed away with it leaves free spaces needed for further modification with an AgNO_3_ solution.

### 2.3. Synthesis of Silver Nanoparticles (AgNPs)

The two series of fibres (A0.0 and B0.0) obtained were saturated with AgNO_3_ solutions of 0.2%, 1% and 5% against the weight of the fibres. The concentrations and determinations of the samples are specified in Table 3.

Figure 2 shows digital images of samples before and after application of theAgNO_3_.

### 2.4. Microbiological Activity

In order to determine the biological activity, non-woven discs were prepared, obtained by pressing chopped fibres using a laboratory press. An example of a photo of nonwoven discs prepared for testing is presented below (Figure 3).

They were exposed to the Gram-positive *Staphylococcus aureus* ATCC No. 25923, Gram-negative *Escherichia coli* ATCC No. 25922, *Pseudomonas aeruginosa* ATCC No. 27853. The microorganisms were purchased from ATCC. The following media were used for the cultivation of microorganisms: Chapman agar, Mac Conkey agar, Cetrymide agar. Microorganisms were incubated at 37 °C, for 24 h. Sterile physiological salt (2 cm^3^) was poured into the cultured bacteria. Grown cultures were washed out with 1 mL of physiological salt solution, and added to the sterile selective agar. The non-woven disc samples (2 cm^2^) were applied to inoculated agars. Samples were incubated at 37 °C, for 24–48 h. The assessment of antibacterial activity consisted of placing a sample on an agar substrate containing bacterial culture and observing its growth under and around the sample. Microbiological tests were repeated for three fibres series and for three types of bacteria, both gram positive and gram negative. The zones of inhibition of microbial growth observed in the conducted studies were determined using an optical microscope.

### 2.5. Methods of Materials Characterization

Scanning electron microscopy (SEM) was used to observe the structure of the samples, which were characterized by JSM 5500 LV made by JEOL (Tokyo, Japan). The microscope was operated in back scattered electron mode, using an accelerating voltage of 10 kV. The samples were gold coated in Jeol JFC 1200 (Tokyo, Japan) ion sputter coater.

The examinations of the surface topography for the samples were observed by an optical microscope (Reichert, Viena, Austria) equipped with an ARTCAM CCD camera (Olympus, Tokyo, Japan), controlled by the Motic Images Plus 2.0 computer program.

The elemental chemical analysis was performed using a Phenom ProX microscope (AM Eindhoven, The Netherlands), with a fully integrated EDS detector and software. The distribution of the different elements in the fibres was evaluated with the element identification (EID) software package and a specially designed and fully integrated energy dispersive spectrometer (EDS).

The spectroscopic investigations were carried out using Nicolet 6700 Fourier Transform spectrophotometer (Thermo Scientific, Waltham, MA, USA) with OMNIC 8.0 software and Easi Diff diffusion accessory (Thermo Scientific, Waltham, MA, USA) was used in the FTIR spectroscopic analysis. The spectral region was as follows: 4000–500cm^−1^, resolution: 4cm^−1^, number of scans: 64 of the solid samples. Each spectrum was analysed with the use of a linear baseline and pre-processed by means of Fourier smoothing.

## 3. Results and Discussion

The digital images present fibres before modification with AgNO_3_ solution, and after modification with AgNO_3_ solutions of different concentrations (Figure 4). As shown in Figure 4, the obtained fibres are dark brown-coloured, which suggests a reduction from Ag^+^ to Ag^0^.

The reaction mechanism of Ag^+^ ion reduction to Ag^0^ atoms which takes place in several stages is schematically shown in the Equations (1)–(4).
(1)$$C_{17}H_{35}{\mathrm{COO}}^{-}{\mathrm{Na}}^{+}\;+\;{\mathrm{Ag}}^{+}{{\mathrm{NO}}_{3}}^{-}\to C_{17}H_{35}{\mathrm{COO}}^{-}{\mathrm{Ag}}^{+}\;+\;{\mathrm{Na}}^{+}{{\mathrm{NO}}_{3}}^{-}$$
(2)$$2C_{17}H_{35}{\mathrm{COO}}^{-}{\mathrm{Ag}}^{+}\;+\;2e^{-}\to 2C_{17}H_{35}{\mathrm{COO}}^{-}\;+\;2{\mathrm{Ag}}^{0}$$
(3)$$C_{3}H_{8}O_{3}\;-\;2e^{-}\to C_{3}H_{6}O_{3}\;+\;2H^{+}$$
(4)$$2C_{17}H_{35}\mathrm{COOAg}\;+\;C_{3}H_{8}O_{3}\to 2C_{17}H_{35}\mathrm{COOH}\;+\;C_{3}H_{6}O_{3}\;+\;2{\mathrm{Ag}}^{0}$$

In the first stage, sodium stearate reacts with AgNO_3_, and silver stearate and sodium nitrate are formed (Equation (1)). The silver ions are reduced by the glycerol located in the fibres and AgNPs are formed (Equations (2), (3) and (4), which penetrate the inside of fibres, where the Ag ions encapsulation. As a result of the reduction, the fibre dyeing characteristic of silver nanoparticles was observed, depending on the concentration of AgNO_3_ solution used.

Scanning electron microscopy enabled the observation of the surface morphology of the obtained fibres. Figure 5 presents SEM micrographs of blends fibres, before and after modification with AgNO_3_ solutions.

The SEM images show a significant difference between the A and B series fibres (Figure 5). The content of sodium stearate in the fibres affects the surface morphology, with a content of 2% sodium stearate, the surface of the fibres is smooth without visible microcracks (series A), which is observed for the addition of 8% stearate (series B). Surface morphology will affect the diffusion of reagents into the fibre. In the case of the A series, the penetration of silver nitrate into the fibres is difficult and the reduction reactions to AgNPs will mainly occur on the surface of the fibre, the observed effect of which are darker fibres (A series). However, in the case of the B series fibres, the resulting microcracks will allow AgNO_3_ to penetrate better into the fibres.

The examination of the surface morphology for the fibres was carried out by optical microscopy (Figure 6).

Microscopic photographs of modified fibres taken in immersion oil show the internal morphology of the fibres. The fibres obtained in the A series have a skin in the form of a continuous outer layer, which prevents the immersion oil from penetrating inside the fibres. The micropores formed in the interior scatter the light, which is revealed in photographs, in the form of dark streaks oriented along the axis of the fibres.

The fibres obtained in the B series have a porous skin—the outer layer, which allows immersion oil to migrate inside the micropores, the effect of which is the weakening of the light scattering through the microchannels, which can be seen in the photographs. At the highest concentration of the additive (5% AgNO_3_), nanoparticles embedded in the fibre microchannels are visible. This type of fibre provides the possibility of as low release of AgNPs from the internal microchannels to the outside, which in effect should extend the bioactivity of the fibres.

EDS tests were performed for samples from the A series with different AgNPs content. Analysis was carried out to confirm the formation of AgNPs. EDS spectra for the elemental analysis were obtained by positioning the laser beam in the centre of randomly selected particles—Figure 7.

The main elements identified in the particles were C, O, Ag, Na (Table 4).

On the spectrograms, no nitrogen was detected, indicating that the reaction is the reduction of Ag^+^ to AgNPs and the removal of NO_3_^−^ groups.

The chemical structures of the EVOH/EPC/glycerol/sodium stearate blends of fibres were studied by infrared spectroscopy. Some feature wavenumbers of EVOH, EPC, glycerol, sodium stearate are labelled in Table 5.

The spectra of all the components of forming fibres are shown in Figure 8.

Figure 8 shows the FTIR spectrum of EVOH (a), glycerol (b), sodium stearate (c) and EPC (d), with the characteristic oscillation bands indicated. A characteristic feature of EVOH and glycerol spectra is the stretching band of O–H oscillators, with a maximum at approximately 3350 cm^−1^. According to literature data [11,38], this is evidence of the presence of strong hydrogen bonds in both components. In EVOH, both the deformation bands of the –C–H oscillators in the –CH_2_–groups at approximately 1440 cm^−1^ and 1330 cm^−1^,as well as the –C–O– oscillators in the –CH–OH– groups at about 1105 cm^−1^, indicate a different configuration within these groups, which is visible in the form of blur and widening of these bands in relation to glycerine or thermoplastic elastomer. Thus, EVOH is a typical semicrystalline polymer.

In the spectra (Figure 9) of fibres from blends of EVOH/EPC/sodium stearate/glycerol with a 2% (spectrum “e”) and 8% (spectrum “d”) content of sodium stearate, a distinct local maximum for the O–H oscillator is characteristic at about 3500 cm^−1^. This is a significant shift in relation to the spectra obtained from the sum of the spectra of the starting components (spectrum “b” and “c”), for which the local maximum for O–H oscillators is located at approximately 3350 cm^−1^. This effect indicates the breaking or loosening of some of the hydrogen bonds present in the fibre material. At the same time, the O–H oscillator band extends significantly towards lower values of wave numbers. This phenomenon can be explained by the presence of strong local interactions between the –OH groups of glycerol, –COONa groups of sodium stearate and –OH groups in EVOH. It is probable that the EVOH chains are spaced apart and some of the O–H oscillators are released from intermolecular hydrogen interactions, which results in shifting their resonance frequencies towards the higher wavenumbers. At the same time, the shift to the fibres’ spectrum, in relation to sodium stearate spectrum, of C=O resonance oscillations in the –COONa group can be observed at 1576 cm^−1^ to 1558 cm^−1^. This proves strong interactions of this group with –OH groups of glycerol or EVOH. Sodium stearate and glycerol are therefore involved in the swelling of EVOH fibres.

The removal of glycerol (comparison in Figure 10) from fibres results in the partial reconstruction of hydrogen bonds in the EVOH material, which causes the local maxima of the O–H oscillator band to shift from 3500 cm^−1^ to approx. 3450 cm^−1^, as well as release from strong hydrogen interactions with glycerol and local formation of a band at about 3175 cm^−1^. Sodium stearate, on the other hand, after the removal of glycerol, is probably precipitated in the form of micelles (agglomerates) in the microporous voids in the EVOH material. The effects of this phenomenon are visible, both under the electron and optical microscope in the form of gaps or cracks arranged along the fibre axis. Confirmation of agglomeration of sodium stearate is a clear reduction in the intensity of the C=O band at about 1560 cm^−1^ in the –COONa group. It is probable that the groups are directed towards –OH groups in EVOH.

Comparison in Figure 11 of A series spectra after saturation with AgNO_3_ solution and reduction of silver ions to AgNPs indicates the progressive process of formation of sodium stearate salt. The C=O oscillators in the –COONa group with the absorption maximum at 1560 cm^−1^ change the resonance frequency, and in the –COOAg groups are in the maximum at 1541 cm^−1^. The formation of complexes around Ag^+^ ions causes a further shift in the maximum of C=O vibrations in the –COOAg group up to 1517 cm^−1^. The intensity of the band at 1647 cm^−1^, characteristic for the deformative vibrations of the –OH oscillator, increases in the –COOH group in stearic acid.

The subsequent stages of the interactions of sodium stearate with the fibre material and its participation in the exchange reaction is shown in Figure 12. The C=O band in sodium stearate (“d” spectrum) is visible at 1576 cm^−1^, then the interaction of the –COONa polar group with –OH groups in glycerol and EVOH and the C=O band shift to approx. 1560 cm^−1^ (“a” spectrum). The exchange reaction of Na^+^ ions to Ag^+^ ions and further shift of wavenumbers for the C=O oscillator in –COOAg to about 1541 cm^−1^ (spectrum “c” Figure 12), followed by surrounding with ligands and the further reduction of the wave number of the C=O oscillator to approx. 1517 cm^−1^.

Antimicrobial activities of AgNPs-fabrics were tested against Gram-negative bacteria of *Escherichia coli (E.coli), Pseudomonas aeruginosa (P. aeruginosa)* and Gram-positive bacteria of *Staphylococcus aureus* (*S. aureus).*

The photographs were taken using an optical microscope. A zone of growth inhibition for individual microorganisms was marked on each of the photographs (Figure 13). The fibre samples with the lowest AgNPs content (A0.2) and the highest AgNPs content (A5.0) were chosen for the appropriate visualization of the effects. For each fibre sample, the tests were repeated for three inoculations.

The determined growth inhibition zones for the bacteria are presented graphically below (Figure 14).

The mean values of growth inhibition zones for all microorganisms of each series were also calculated. The results are presented in Table 6. Mean values were calculated for zones of inhibition for each of the three series.

Based on the conducted microbiological tests, it was found that the obtained fibres after modification of AgNPs showed bioactivity against all bacteria used in the research. It was found that the size of the microorganism growth inhibition zones increased with the amount of AgNPs nanoparticles content in the fibres.

The fibres obtained in series A have higher zones of inhibition compared to the fibres of series B for Gram-negative bacteria used in the test. This could be due to the higher amount of AgNPs nanoparticles that may accumulate on the surface of the fibres as they contained less sodium stearate. On the other hand, the fibres obtained in the B series have higher inhibition zones compared to the A series fibres for the gram-positive bacteria used in the test. This may be due to the fact that the composition of the mixture used for the preparation of fibres contained four times more sodium stearate, which influenced the morphology of the obtained fibres. The obtained fibres are more porous, and have many microchannels compared to the fibres obtained in the A series. Probably due to the structure obtained in this way, more silver ions could penetrate and occlude inside the fibres. The control samples, which were fibres untreated with AgNO_3_ solutions, showed no signs of bioactivity towards the microorganisms used.

## 4. Conclusions

In this study, we prepared bioactive fibres which had a microporous structure with EVOH blend/EPC applications agents. In the performed tests, various additive compositions were used, which significantly influenced the structure and properties of the new fibres. A significant change in fiber color was observed depending on the AgNO_3_ concentration. Modification of AgNO_3_, followed by reduction to AgNP_S_, made the fibres bioactive. Bioactivity has been confirmed for various types of Gram-positive and Gram-negative bacteria. Higher bacterial activity was confirmed for gram positive bacteria. The obtained new bioactive fibres can be used for the production of modern medical devices, e.g., protective clothing, including protective masks. EVOH is a polymer with hydrophilic properties; additionally, the application of silver has a positive effect on the extension of application properties.

## Figures and Tables

**Figure 1 polymers-12-01827-f001:**
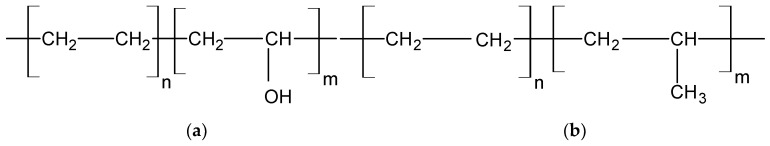
The structure of components: (**a**) EVOH, (**b**) EPC, (**c**) glycerol, (**d**) sodium stearate.

**Figure 2 polymers-12-01827-f002:**
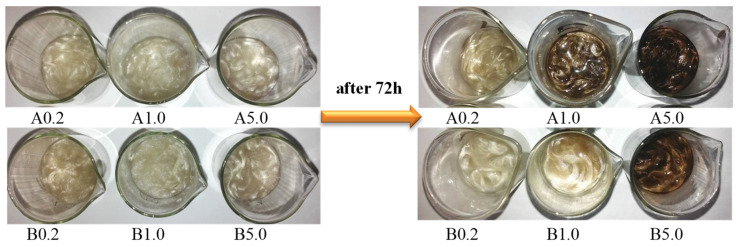
The samples before and after 72h to application of theAgNO_3_.

**Figure 3 polymers-12-01827-f003:**
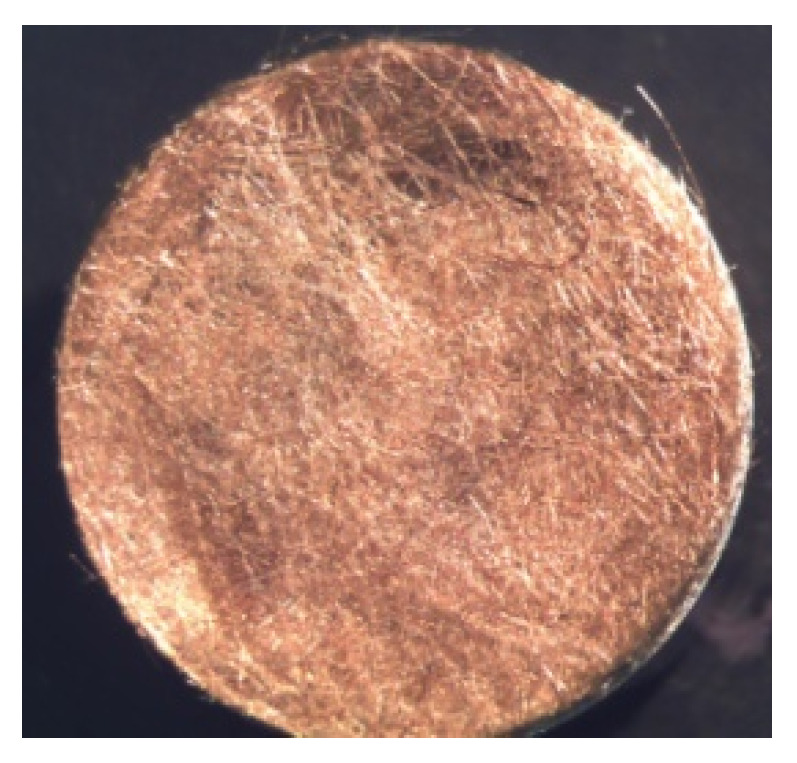
Digital images of nonwoven disc.

**Figure 4 polymers-12-01827-f004:**
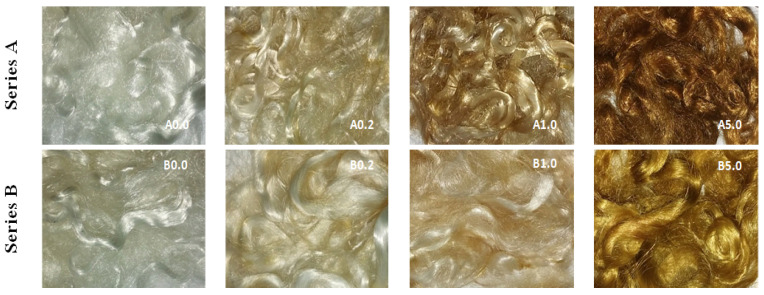
Digital images of fibres before and after application of theAgNO_3_.

**Figure 5 polymers-12-01827-f005:**
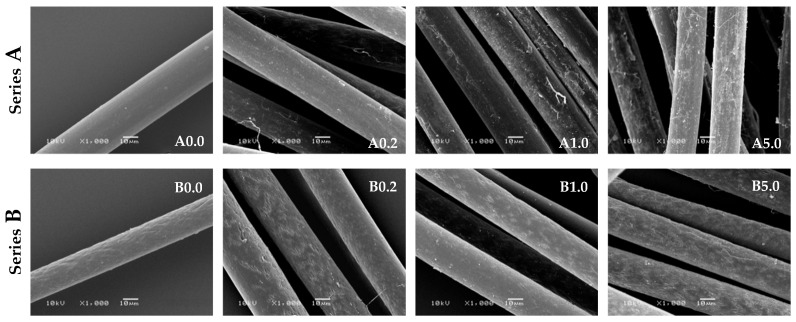
SEM micrographs of blends fibres before and after modification with various concentrations of AgNO_3._

**Figure 6 polymers-12-01827-f006:**
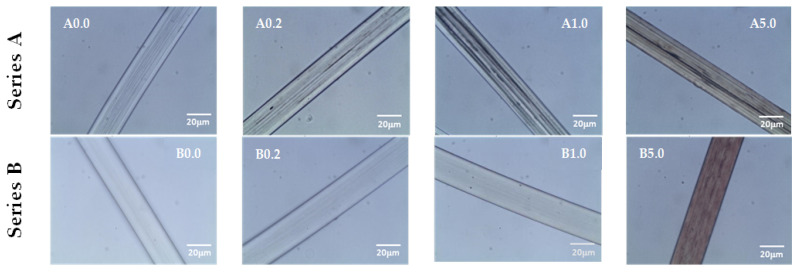
Optical microscopy images of fibres.

**Figure 7 polymers-12-01827-f007:**
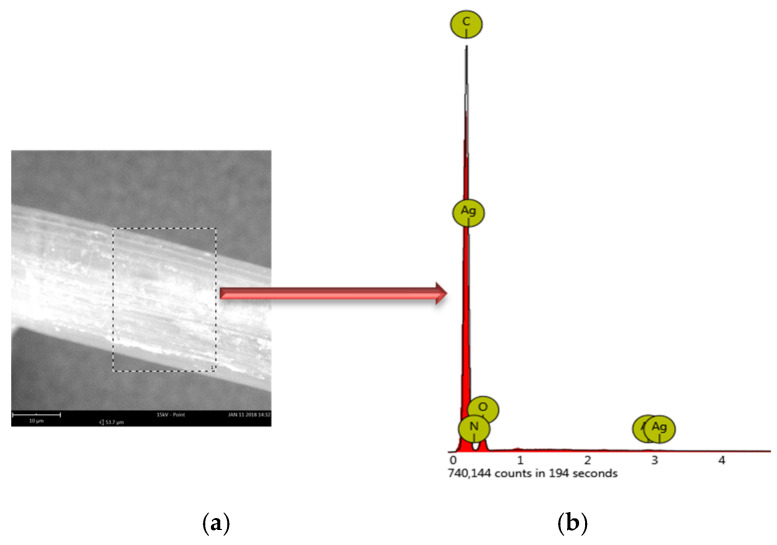
(**a**) SEM and surface mapping (the dotted line presents the path of EDS line scan of samples A0.2), (**b**) EDS analysis of particles (A0.2).

**Figure 8 polymers-12-01827-f008:**
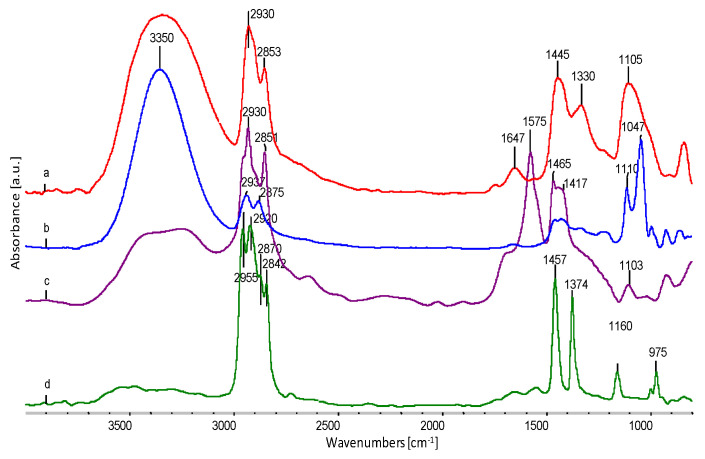
FTIR spectra of fibre forming components used; (**a**) EVOH, (**b**) glycerol, (**c**) sodium stearate, (**d**) EPC with the characteristic oscillation bands.

**Figure 9 polymers-12-01827-f009:**
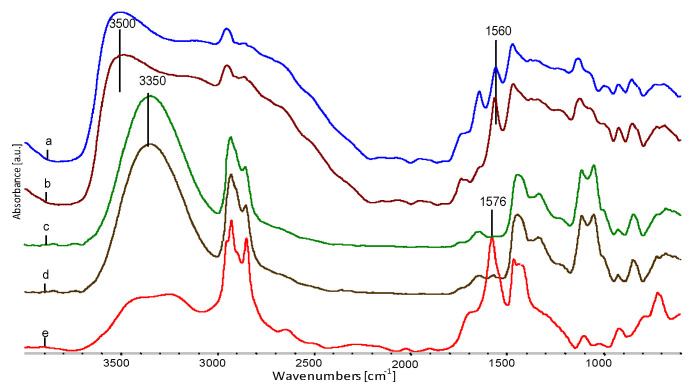
FTIR spectra of fibre spectra obtained by adding spectra of the following components: (**a**) A series, non-extracted; (**b**) B series, non-extracted; (**c**) A series, obtained as a result of the mathematical treatment of the percentage of components; (**d**) B series obtained as a result of the mathematical treatment of the percentage of components; (**e**) spectrum for sodium stearate.

**Figure 10 polymers-12-01827-f010:**
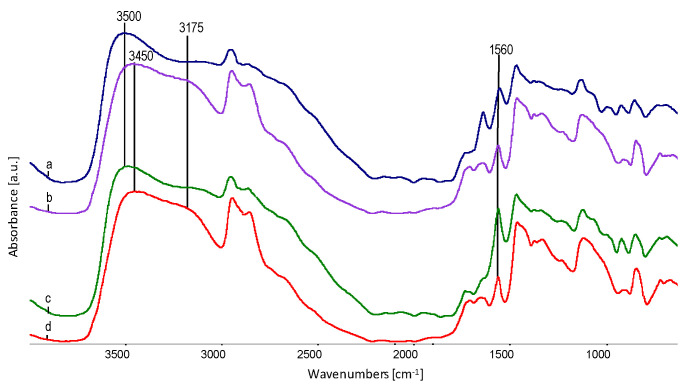
Combination of fibre spectra before and after extraction in water: (**a**) A series not extracted; (**b**) A series extracted; (**c**) B series not extracted; (**d**) B series extracted.

**Figure 11 polymers-12-01827-f011:**
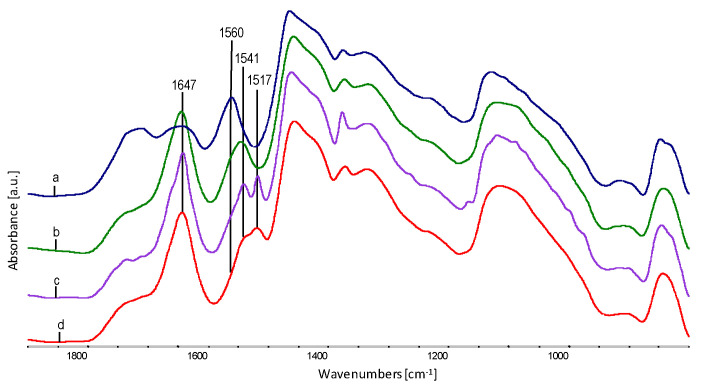
FTIR spectra within a range of 1900–800 cm^−1^ of fibre spectra, after extraction in water and AgNO_3_ saturated A-series fibres and reduced to AgNPs: (**a**) spectrum for A0.0; (**b**) spectrum for A0.2; (**c**) spectrum for A1.0; (**d**) spectrum for A5.0.

**Figure 12 polymers-12-01827-f012:**
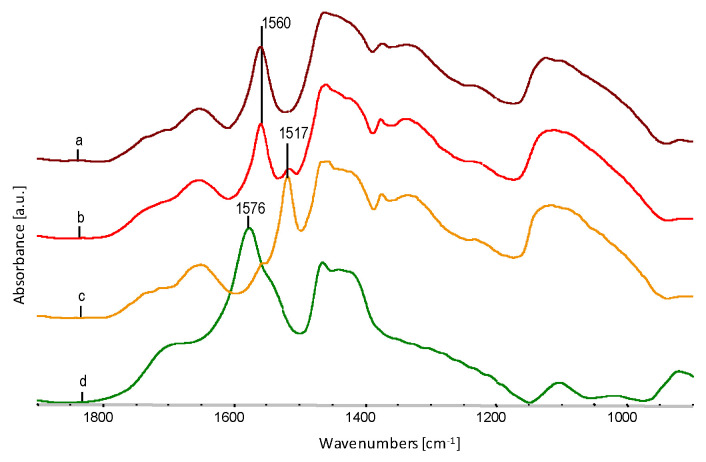
Comparison of spectra of B series fibres saturated with AgNO_3_ and reduced to AgNPs and after extraction in water and sodium stearate: (**a**) spectrum for B0.2, (**b**) spectrum for B1.0, (**c**) spectrum for B5.0, (**d**) spectrum for sodium stearate.

**Figure 13 polymers-12-01827-f013:**
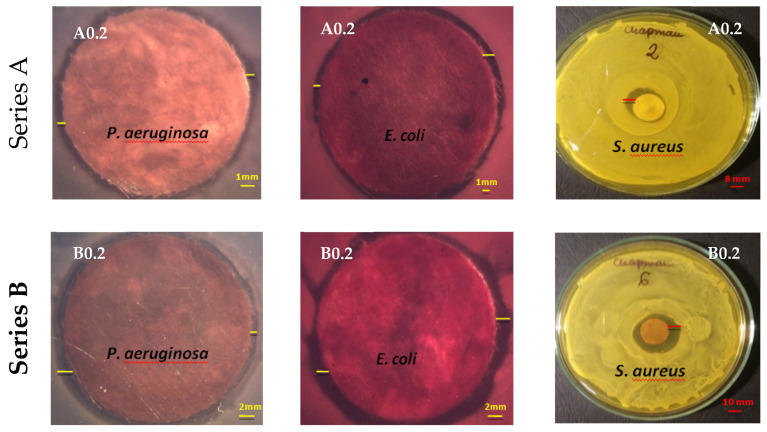
Representative images of the agar plates showing the diameter of the inhibition zone for *P. aeruginosa*, *E. coli* and *S.aureus*.

**Figure 14 polymers-12-01827-f014:**
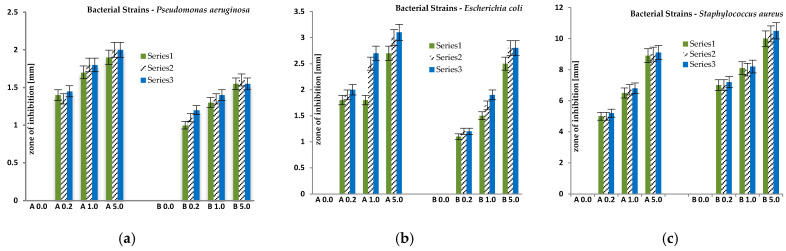
Inhibition zone assay of blends of fibres against *P. aeruginosa* (**a**), *E. coli* (**b**) and *S. aureus* (**c**).

**Table 1 polymers-12-01827-t001:** Properties of ethylene–vinyl alcohol copolymers (EVOH) and ethylene-propylene copolymers (EPC).

Properties	EVOH	EPC
Ethylene content [mol%]	32	13
Density [g/cm^3^]	1.19	0.865
Melting temperature [°C]	180	-
Crystallization temperature [°C]	155	-
Melt Flow Index (MFI) (190 °C, 2160 g) [g/10 min]	1.51	21

**Table 2 polymers-12-01827-t002:** Compositions of the prepared EVOH/EPC/glycerol/sodium stearate blends.

Component *w/w* [%]	Sample A0.0	Sample B0.0
EVOH	57	57
EPC	3	3
glycerol	38	32
sodium stearate	2	8

**Table 3 polymers-12-01827-t003:** The applied concentrations and designations of samples in this study.

Samples for Treatment	Concentrations AgNO_3_ [%]	Designations of Samples
A0.0	0.2	A0.2
A0.0	1.0	A1.0
A0.0	5.0	A5.0
B0.0	0.2	B0.2
B0.0	1.0	B1.0
B0.0	5.0	B5.0

**Table 4 polymers-12-01827-t004:** EDS analysis of samples A0.2 and A1.0.

Symbols of Elements	Atomic Percent A0.2	Atomic Percent A1.0
C	80.82	75.44
O	19.03	23.66
Ag	0.10	0.80
Na	0.05	0.10
N	0.00	0.00

**Table 5 polymers-12-01827-t005:** Wavenumbers of the bands observed in the FTIR spectra of EVOH, EPC, glycerol, sodium stearate bands [11,37].

Sample	Wavenumber from FTIR, cm^−1^	Oscillation Bands
EVOH	3350	O–H str. in hydrogen bond
	2930	CH_2_ str. asym.
	2853	CH_2_ str. sym.
	1647	O–H def.
	1445	CH_2_ sc.
	1330	CH_2_ def.
	1105	C–O II-secondary
EPC	2955	CH_3_ str. asym.
	2920	CH_3_ str.sym.
	2870	CH_2_ str. asym.
	2842	CH_2_ str.sym.
	1457	CH_2_ sc.
	1374	CH_2_ def.
	1160	C–C str.
	975	CH_3_ roc.
Glycerol	3350	O–H str. in hydrogen bond
	2937	CH_2_ str. asym.
	2875	CH_2_ str.sym.
	1110	C–O II-secondary
	1047	C–O I-primary
Sodium stearate	2930	CH_2_ str. asym.
	2851	CH_2_ str.sym.
	1576	C=O in COONa
	1465	CH_2_ sc.
	1417	CH_3_ sc.
	1003	C–C sk.

**Table 6 polymers-12-01827-t006:** Zone of inhibition for various sample treatments against Gram-negative *Escherichia coli, Pseudomonas aeruginosa* and Gram-positive *Staphylococcus aureus* from an average of three different experiments, with a standard deviation of ±0.1–0.5 mm.

Samples	Average Zone of Inhibition ± 0.1 ÷ 0.5 [mm]		
	*Pseudomonas**aeruginosa* Gram-Negative	*Escherichia coli* Gram-Negative	*Staphylococcus aureus* Gram-Positive
A 0.0	-	-	-
A 0.2	1.4	1.9	5.0
A 1.0	1.8	2.3	6.7
A 5.0	2.0	3.0	3.0
Average zone of inhibition [mm]	1.7	2.4	4.9
B 0.0	-	-	-
B 0.2	1.1	1.2	7.1
B 1.0	1.4	1.7	8.1
B 5.0	1.6	2.7	10.3
Average zone of inhibition [mm]	1.4	1.9	8.5

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
