# Peer review of "Preparation of Antimicrobial Fibres from the EVOH/EPC Blend Containing Silver Nanoparticles"

_polymers, 2020, doi:10.3390/polym12081827_

Round 1

Reviewer 1 Report

This manuscript presents good research work related modification of fiber morphology with sodium stearate and its effect on the microbiological activity of the fabricated fiber treated with AgNPs solution. The authors analyzed the chemical and morphological change systematically.  Thus I recommend this paper to be published in Polymers with minor revision. Some minor comments are as follows.

  1. Please describe the main purpose of this study in the Introduction section. It the main goal is to increase the antibacterial durability by including more silver particles inside the fiber by surface treatment, it will be necessary to provide the maintenance of antimicrobial properties after multiple washing. 
  2. Provide the average size of AgNPs prepared and average pore size generated on the surface of fiber
  3. Please state clearly the main conclusions and provide an explanation of the importance and relevance of the study to the field.

Reviewer 2 Report

The present topic is good, but how they say it's a novel and Ag formed as a nano, I hope that they should present clearly and need to present a nano-related characterization of this

Author Response

Thank you for your review.

The reviewer rightly noted the lack of direct evidence of the nano size of Ag particles. The main objective of the research work undertaken was to form microporous fibers based on EVOH / ETP modified with AgNPs and to determine the influence of the additives used on the structure and properties of the resulting innovative fibers. Microporous fibers with a different surface structure and interior of the fiber core were obtained. The resulting micropores are connected to nanochannels inside the fiber, invisible under an optical microscope. The filling of micropores and nanochannels with silver particles should be related to the nano-size of Ag particles formed inside. This effect is observed in images taken from an optical microscope [Fig.6]. It is also confirmed by their bioactivity.

Round 2

Reviewer 2 Report

I hope the present journal will be acceptable